# Transient Fluorescence Labeling: Low Affinity—High Benefits

**DOI:** 10.3390/ijms222111799

**Published:** 2021-10-30

**Authors:** Maxim M. Perfilov, Alexey S. Gavrikov, Konstantin A. Lukyanov, Alexander S. Mishin

**Affiliations:** Shemyakin-Ovchinnikov Institute of Bioorganic Chemistry, Russian Academy of Sciences, 117997 Moscow, Russia; sufrep@gmail.com (M.M.P.); gavrikovalexey1@gmail.com (A.S.G.); kluk@ibch.ru (K.A.L.)

**Keywords:** super-resolution microscopy, PAINT, fluorescent labeling, exchangeable labels

## Abstract

Fluorescent labeling is an established method for visualizing cellular structures and dynamics. The fundamental diffraction limit in image resolution was recently bypassed with the development of super-resolution microscopy. Notably, both localization microscopy and stimulated emission depletion (STED) microscopy impose tight restrictions on the physico-chemical properties of labels. One of them—the requirement for high photostability—can be satisfied by transiently interacting labels: a constant supply of transient labels from a medium replenishes the loss in the signal caused by photobleaching. Moreover, exchangeable tags are less likely to hinder the intrinsic dynamics and cellular functions of labeled molecules. Low-affinity labels may be used both for fixed and living cells in a range of nanoscopy modalities. Nevertheless, the design of optimal labeling and imaging protocols with these novel tags remains tricky. In this review, we highlight the pros and cons of a wide variety of transiently interacting labels. We further discuss the state of the art and future perspectives of low-affinity labeling methods.

## 1. Introduction

Most fluorescent labeling methods are based on the permanent—high-affinity or covalent—attachment of dyes to the target molecules. For example, immunofluorescence techniques [1] employ fluorescently labeled antibodies to directly or indirectly stain proteins or other antigens. Another popular approach is the introduction of a genetically encoded tag into the structure of the target protein. Such fusion proteins should be expressed in the cells of interest in culture or in vivo. Green fluorescent protein (GFP) and homologous fluorescent proteins (FPs) represent self-sufficient tags, which develop fluorescence on their own shortly after protein synthesis due to the formation of a fluorophore inside the FP barrel [2]. There are also other classes of fluorescent proteins that fluoresce due to the binding of endogenous cofactors such as flavins or biliverdin [3]. These tags are convenient but have a limited color palette. Finally, a diverse set of exogenously applied chemical dyes can be permanently attached to the protein of interest fused with specially designed genetically encoded enzymatic tags, such as SNAP-tag and HaloTag (reviewed in [4]).

Still, every method of permanent labeling has its own limitations. For example, antibodies-derived labels are highly specific to target structure [5,6] and allow using various organic dyes conjugated with antibodies. However, their field of application is usually limited to fixed cells. Furthermore, their sheer size (>10 nm [7]) may increase the apparent dimensions of a labeled structure, especially in the case of a combination of primary and secondary antibodies [7]. In multitarget imaging, the use of multiple antibodies could lead to spatial interference between the antibodies and mislocalization of targets [8]. Ultimately, the size of antibodies limits the effective labeling density [8]. 

Another covalent labeling technique is genetically fusing fluorescent proteins [9]. Labeling with FPs remains at present the method of choice for live-cell microscopy of intracellular protein targets, although some FPs could be effectively used in fixed cells as well [10]. With the smaller size of the FP-tags, higher sampling density can be achieved, in comparison with antibody-based labeling. Indeed, the sizes of an actin subunit and fluorescent proteins are close (≈6 [11] and ≈5 nm [12] in max projection, respectively), which in theory makes it possible to label each monomer. However, in practice, cells express their own unlabeled actin, which also incorporates into a microfilament, thus diluting the stained monomers and ultimately lowering the labeling density [13]. It is curious that despite its relatively small size, FP-fusion nevertheless is bulky enough to perturb the functionality of the target proteins [14]. In addition, fluorescent proteins are strongly susceptible to damage caused by light irradiation. This process is called photobleaching, and it significantly reduces the span of time available for continuous imaging with FP [15]. Susceptibility to photobleaching, incomplete photoconversion [16], and incomplete maturation make fluorescent proteins less-than-optimal for some types of super-resolution microscopy [7].

Finally, labeling is possible by organic dyes, conjugated with small molecules responsible for homing to target structures such as fluorescently labeled phalloidin for actin staining [17] or 4,4-difluoro-4-bora-3a,4a-diaza-s-indacene (BODIPY) for membrane staining [18]. Additionally, silicon-rhodamine conjugates SiR-actin (with desbromo-desmethyl-jasplakinolide) and SiR-tubulin (with docetaxel) stain actin filaments and microtubules, respectively [19]. Although these probes demonstrate bright and photostable labeling in various types of microscopies, both in vitro and in vivo, they slightly affect the polymerization and functionality of actin or tubulin, making the interpretation of live-cell imaging with these dyes more complicated [19,20].

The effective alternative to labels with high affinity to target could be low-affinity tags (with K_D_ in the micromolar range). Kiuchi et al., demonstrated that exchangeable probes provide high labeling density [8]. In addition, the labeling becomes more photostable [21] and less toxic and disruptive to the functioning of cellular structures [14,22]. The basic advantages of exchangeable probes over high-affinity-based ones are presented in Figure 1.

Low-affinity labeling is a diverse group that includes such methods as PAINT (point accumulation for imaging in nanoscale topography) for membrane imaging, fluorogen-activating proteins, and exchangeable organic dyes of a narrow specificity. Therefore, it could be challenging to choose a suitable method for each particular experiment. In an attempt to make this task easier, herein, we consider and compare low-affinity labeling techniques, highlight the general principles underlying such techniques, and suggest future directions of development.

## 2. PAINTing the Cell

A large family of methods utilizing exchangeable probes shares ‘PAINT’ in their names. Prerequisites for the development of PAINT were developed in the work of Mei et al. [23]. The authors used the fluorogenic property of the Nile Red probe: in the hydrophobic environment, it exhibits a much higher quantum yield than in the hydrophilic one [24]. This feature allowed observing the probe only at the moment of interaction with target hydrophobic vesicles. In this seminal work, an unusual technique, trajectory time distribution optical microscopy (TTDOM) was used, so instead of measuring the intensity of light, the statistics of probe–target collisions were determined. TTDOM allowed for high-resolution imaging [25], but it required complex and hard-to-reach equipment, preventing the mass adoption of this technique.

In contrast, the PAINT method could be implemented with a standard microscopy gear [26]. With the very same Nile Red probe, Sharonov and Hochstrasser demonstrated precise imaging of large unilamellar vesicles. In essence, PAINT is continuous imaging of the sample, in the presence of a mobile fluorescent probe, which constantly binds and unbinds the target structure. Binding events result in bursts of fluorescence, followed by unbinding of probes and loss of signal. Importantly, the probe concentration controls the collision rate and labeling density. The recommended density of about one molecule per μm^2^ ensures that fluorescent burst corresponds to point spread function (PSF) from a single molecule, allowing for precise localization. Another essential set of parameters is the on-time 𝝉_on_ of fluorescent burst (or “bound” time 𝝉_b_) and the time between collisions 𝝉_off_ (or “dissociated” time 𝝉_d_). For better results, the time between frames should not exceed the former, and the latter should be longer than the exposure time of the frame. The compliance with the specified conditions enables frame-by-frame determination of fluorophore coordinates, which could be summed to the final reconstructed image, similar to other localization microscopy implementations (Figure 2A) [27,28,29].

Later, with the invention of DNA-PAINT [30], the PAINT principle was applied for the labeling of nucleic acids and proteins. The central imaging principle was borrowed from single-molecule speckle microscopy: only immobilized probes give rise to discrete fluorescent signals, while unbound probes remain undetectable [32]. In DNA-PAINT, the DNA-origami structures [33] with docking strands (“docking”) imaged with 7–9 nt long oligonucleotide strands conjugated with dyes (“imager”) [30]. The transient interaction between the “docking” and the “imager” was followed by a fluorescent burst lasting for 𝝉_b_, while the dissociation caused a lowering of the fluorescent signal for 𝝉_d_ (Figure 2B). Moreover, the length of the imager’s strand could be additionally tuned to achieve the optimal combination of 𝝉_b_ and 𝝉_d_. The combination of specific exchanging labels and TIRF (total internal reflection fluorescence) microscopy, which reduces the out-of-focus signal from unbound dyes, provided a reasonable signal-to-noise ratio of images. In addition to imaging, DNA-PAINT was used in the qualitative characterization of DNA-origami structures [34,35].

In turn, protein imaging required additional adapters between DNA strands and target proteins, such as antibodies [34] or aptamers [36]. Based on DNA-PAINT with antibodies, several approaches were developed. Among them were the method for the quantification of target molecules (qPAINT [34]) and Exchange-PAINT for multitarget imaging using orthogonal pairs of strands [37,38]. A DNA-PAINT variant tPAINT was used for live-cell dynamic tension imaging [39], with the help of an additional stem-loop structure included within the complex of interacting DNA molecules, designed to expose a cryptic docking site under external force.

One more implementation of DNA-PAINT combined it with yet another method for improving the resolution—super-resolution optical fluctuation imaging (SOFI) [40]. In SOFI, the reconstruction of super-resolved images is based on the analysis of temporal fluorescence fluctuation of fluorophores. Although the SOFI was presented as a fast method for contrast-enhanced and background-reduced imaging [40], today STORM (stochastic optical reconstruction microscopy) and PALM (photoactivated localization microscopy) methods could be used for high temporal resolution imaging [21,41] with much better spatial resolution of reconstructed images [42].

Thus, the remaining benefits of SOFI are in the absence of specific requirements for labels such as photo-switching, and the tolerance of the method for densely labeled targets [40,43]. Still, the usability of dyes and fluorescent proteins is fundamentally limited by their photobleaching. To overcome this limitation, Glogger et al., combined SOFI with the DNA-PAINT method [44]. As a result, a high concentration of imager strand compared to original DNA-PAINT [30] generated sufficient target-specific fluctuations and allowed an increase in both contrast and resolution.

It is worth considering separately the uPAINT approach: an exception from transient labeling in the PAINT family. The “uPAINT” stands for “universal PAINT” due to its usability for various biomembrane molecules with specific ligands conjugated with fluorophores [31]. The authors used low-angle (or grazing angle) epi-illumination to filter out unbound probes, allowing the excitation of relatively thin layers (Figure 2C) [45]. However, unlike other PAINT methods, uPAINT does not require exchangeable labels.

Finally, it is possible to image RNA molecules in a PAINT-like manner. RNA staining with non-covalent binding of fluorogenic dyes, pioneered in Spinach aptamer [46], have developed to a mature and extensive toolset (reviewed in [47]). One recent example, RhoBAST (rhodamine-binding aptamer for super-resolution imaging techniques) [48], utilizes the main ideas of original PAINT—transient interactions and fluorescence activation upon probe-target binding. Like DNA-PAINT methods, RhoBAST exploits ligand exchange, but with much faster kinetics.

## 3. Fluorogen-Activating Proteins (Protein-PAINT)

In another approach to transient labeling, developed in parallel with DNA-PAINT, a specially designed protein reversibly binds a small molecule of fluorogen, activating (turning “ON”) its fluorescence. Within the protein–fluorogen complex, the fluorescence of the fluorogen can be enhanced mainly by limiting the conformational mobility, the difference in the polarity of the amino acid microenvironment within the binding pocket, or binding-induced changes of the protonation state. In the literature, the proteins in these protein–fluorogen complexes are dubbed fluorogen-activating proteins (FAPs).

In one of the first works in the field of protein–fluorogen interactions, a molecule from the class of so-called fluorescent molecular rotors was used, which binds to polymerized tubulin in vitro, resulting in limitation of fluorogen rotational relaxation and fluorescence enhancement [49]. Another research group obtained monoclonal antibodies specific to a fluorescent molecular rotor, resulting in a more than 40x fold increase in fluorescence quantum yield [50]. While the principle of non-covalent binding of fluorogen and its exchangeability has not yet been exploited in this work, the manuscript underpins the much later antibody-based labeling system.

In 2008, Szent-Gyorgyi et al., isolated human single-chain antibodies (scFv) specific to thiazole orange and malachite green derivatives (Figure 3) and used them to visualize cell surface and secretory apparatus [51]. This labeling system based on the binding of fluorogens to antibodies developed rapidly. A broad palette of antibody variants and fluorogenic dyes appeared, covering almost the entire visible spectrum from blue to near-infrared [52,53]. The use of such antibodies has recently been expanded: a photosensitizer [54], a pH sensor [55], and the so-called affibodies [56,57,58] were created. The affinity of most of these complexes of fluorogenic dyes with antibodies is relatively strong and lies in the low nanomolar range [51]. This labeling system can also be used for single-molecule localization microscopy (SMLM) both in living and fixed cells, showing good reconstruction quality and localization density during acquisition [17].

Yet another representative of fluorogen-activating proteins is the UnaG protein, which reversibly binds the endogenous ligand bilirubin (Figure 3) and activates its fluorescence [59]. This protein was found to be responsible for the fluorescence of the skeletal muscles of the Japanese eel [59,60]. UnaG localizes mainly in the small-diameter muscle fibers of eels. This protein is smaller than conventional fluorescent proteins, about 15.6 kDa, and binds bilirubin tightly (K_D_ = 98 pM). However, the bilirubin–UnaG complex can be photo-switched to a non-fluorescent state through bilirubin’s interaction with oxygen. Due to the non-covalent binding of bilirubin with UnaG, oxidized bilirubin detaches from the protein, freeing up the binding site for another bilirubin molecule [61]. The kinetics of this process is easily controlled by the illumination intensity and the concentration of the ligand. The exact dissociation constant for oxidized bilirubin remains unknown. Nevertheless, UnaG has been successfully used in single-molecule localization microscopy [61].

Later, a yellow fluorescence-activating and absorption-shifting tag (Y-FAST) was developed [62]. Fluorogen-activating protein FAST, which is almost two times smaller than conventional fluorescent proteins (≈14 kDa), was obtained by directed evolution of bacterial photoactive yellow protein PYP. A unique feature of the FAST system is a high signal-to-background ratio, caused by the bathochromic shift of the excitation spectra of the fluorogen within the protein complex due to its deprotonation upon binding. The first fluorogen developed for FAST was 4-hydroxy-3-methylbenzylidene rhodanine (HMBR), which resembles the GFP chromophore. In a complex with Y-FAST, the excitation peak of HMBR (Figure 3) is shifted to 481 nm with the emission peak at 540 nm. In contrast, unbound HMBR remains fully protonated in the physiological range of pH. Despite low affinity (K_D_ ≈ 130 nM, corresponding to the residence time in the complex of 160 ms at 25 °C) [62], a high signal-to-background ratio allows for successful labeling of target cellular proteins. Y-FAST quickly started to develop, and a set of useful chemical-genetic tools based on FAST protein appeared. For example, the iFAST mutant and its tandem variant td-iFAST with improved brightness and many red fluorogens were created [63]. The variety of colors and exchangeability of fluorogens made it possible to perform multicolor dynamic labeling of proteins in living cells by alternately washing and adding fluorogens [64,65,66,67]. Since then, a split system based on FAST has appeared [68], as well as orthogonal reporters greenFAST and redFAST [69], and new, improved FAST mutants with novel fluorogens [70]. Recently, following NMR analysis of the FAST-fluorogen complex, a shortened nanoFAST tag only 98 amino acid residues long was developed [71].

FAST has also been applied to single-molecule localization microscopy in living and fixed cells. Due to the exchangeability and activation of fluorescence, it is possible to detect fluorogen binding events as bursts of fluorescence [72]. However, the reconstruction of FAST-labeled cell structures in living cells was quite tricky. The fluorogen was not photostable and, at low concentrations, quickly photobleached under high laser power illumination. As a remedy, a special buffer with oxygen scavengers was used in fixed cells, prolonging the acquisition. However, it took about 1 h and 90,000 frames to reconstruct the microtubules due to low localization density. Despite the dubious applicability of FAST in classical implementations of single-molecule localization microscopy, the exchangeability of the fluorogen provides local fluctuations in the fluorescence intensity. This property made it possible to perform super-resolution radial fluctuations (SRRF) analysis [73].

A short time after the appearance of the labeling system based on FAST, a similar system called DiB (dye in Blc) appeared with a fluorogen-activating protein based on bacterial lipocalin Blc and fluorogens—analogs of the GFP chromophore (Figure 3) [74]. The development of DiB tags was guided by in silico mutagenesis and molecular docking of the GFP chromophore. Then, the best mutants (further: DiB1, DiB2, and DiB3) were screened in vitro against a fluorogens library. One of the best-performing fluorogens was M739 with the distinctive feature that it contains a fluoroborate group which blocks photoisomerization, therefore increasing the fluorescence quantum yield. The dissociation constants of DiBs complexes with M739 ranged from 0.1 to 9 μM. In terms of photostability, they are superior to conventional fluorescent proteins and can be used in single-molecule localization microscopy, showing a high density and stability of the number of localizations. This makes it possible to reconstruct the labeled structures in a reasonably short time in high quality. The photostability of the DiBs is also beneficial for light-intensive super-resolution approaches, such as stimulated emission depletion, where DiBs outperformed fluorescent proteins. Later, a red fluorogen compatible with DiBs was published, which was also applicable for localization microscopy in living cells [75]. In addition, a self-assembling split system was created based on the DiB2 scaffold, which, like the full-length DiBs, showed high performance in super-resolution microscopy, and the stability of localization density turned out to be even higher than that of the full-length DiB2 [76].

Crystallization of DiB1 with the fluorogen M739 underpinned rational mutagenesis of all three DiBs, thereby improved variants were created [77]. The DiB3/F74V and DiB3/F53L/F74L/L129M mutants in complex with fluorogen M739 have an emission peak of about 540 nm, while the emission of DiB3/F53L is red-shifted up to 562 nm. The differences in spectra were sufficient for simultaneous two-channel single-molecule localization microscopy of structures labeled by different DiB3 mutants. Furthermore, improved DiB variants exhibit increased stability of localization density, single-molecule brightness, and localization precision. In addition, a temporal resolution of about one super-resolved reconstruction per minute was demonstrated.

Recent advances in computational structural biology allowed for the 69 design of β-barrel-type proteins that bind fluorogens and activate their fluorescence [78]. The dissociation constants of these de novo designed mFAP1 and mFAP2 proteins and the 3,5-difluoro-4-hydroxybenzylidene imidazolinone (DFHBI) fluorogen were 0.18 and 0.56 μM, respectively. Unfortunately, the fluorescence quantum yield of complexes with the chromophore DFHBI (Figure 3) was only about 2%, and in terms of relative brightness, these tags were ≈35 times dimmer than the fluorescent protein EGFP. Despite this, the de novo β-barrel has become a versatile platform for introducing biosensory functions into it [79]. Thus, a split of mFAP was created to detect the association and dissociation of proteins. It was shown that it is possible to design improved variants of mFAPs, one of which, mFAP10, is only two times dimmer than EGFP, reaching a fluorescence quantum yield of about 23% in complex with the 3,5-difluoro-4-hydroxybenzylidene-2,2,2-trifluoroethyl imidazolinone (DFHBI-1T) fluorogen (Figure 3). Biosensors for pH and calcium levels were also designed. Overall, the mFAP platform showcased the current level of computational structural biology, which no doubt will be one of the pillars of the development of novel protein tags.

The most recently developed system for chemogenetic labeling of proteins with exchangeable probes is based on dimeric transcription factors LmrR and RamR [80]. The modified transcription factors reversibly bind commercially available DFHBI and BODIPY chromophores (Figure 3). Dissociation constants ranging from submicromoles to several micromoles allow for simple chromophore washing and staining protocol. The system, dubbed chemogenetic tags with probe exchange (CTPEs), is well suited for in vivo labeling of bacterial proteins. To reduce the affinity of transcription factors to DNA, point mutations were introduced into the DNA binding interface. Interestingly, within one homodimer LamR molecule, there are two pockets for binding chromophore in contrast to RamR, in which there is only one pocket in the homodimer.

Overall, labeling systems with fluorogen-activating proteins has gained some traction in the community. The key advantages of FAPs are the ability to carry out prolonged acquisition or alternate staining in living cells, which cannot be done using conventional fluorescent proteins, irreversibly anchored to the labeled structure. In addition, some systems perform very well in super-resolution microscopy in both fixed and living cells (Table 1).

More than ten years have elapsed since the development of an antibody for a molecular rotor dye before this principle was first used for the fluorescent labeling of proteins. Today, the class of FAP tagging systems is growing quite rapidly, and different niches have emerged in it. Notably, Y-FAST-related labels are well suited for protein labeling in living cells and even whole organisms using wide-field microscopy. It is also possible to carry out simultaneous two-color labeling with orthogonal FAST variants. The very low non-specific chromophore signal in the membranes and the high fluorescence enhancement upon binding make FAST an excellent fluorescent labeling system. The labeling system based on bacterial lipocalin from the DiB family performs well in wide-field microscopy but is inferior to the system based on Y-FAST because of a higher signal of chromophores associated with cell membranes. However, in localization nanoscopy, DiBs have no equal. High stability of localization density in combination with high brightness at the level of single molecules makes DiBs an ideal marker for super-resolution in living cells, including time-lapse nanoscopy. Despite the relatively active development of fluorogen-activating proteins in recent years, each fluorogen–protein pair has its strengths and weaknesses. Therefore, there is still no universal tagging system that could be used in any conditions for any task.

## 4. Cytoskeleton Labeling

One of the most spectacular intracellular structures from the point of view of a microscopist is a cytoskeleton. Usually, it consists of three types of fibers—microfilaments, intermediate filaments, and microtubules. Conventional or high-affinity tags are the most widely used for staining the cytoskeleton. However, nearly all of them exhibit low photostability, low labeling density, or could not be used in live-cell microscopy [7]. In addition, a common feature of cytoskeleton fibers is their composition of monomers which polymerize and depolymerize in response to cellular stimuli. Importantly, some high-affinity labels, such as phalloidin-based labels [82], stabilize the polymerized form, disrupting normal cellular functioning [83,84].

As a way to overcome these limitations, the low-affinity transiently interaction labels were introduced [7,8]. One example is a Lifeact—a short (just 17 amino acids) fragment of actin filament-binding protein Abp140 from *Saccharomyces cerevisiae* [14]. Low affinity to F-actin (K_D_ ≈ 2.2 μM) results in a rapid exchange of probes (within 0.4 s) [14]. Additionally, Lifeact was used as an actin-labeling probe in the labeling technique named IRIS (image reconstruction by integrating exchangeable single-molecule localization) [8]. In IRIS, transient interaction of diverse fluorescently-labeled proteins and their protein partners within the fixed cell sample resulted in staining of target structures. In a manner similar to the single-molecule speckle microscopy [32], binding events visible at low concentration of fluorescently-labeled protein probes were registered as fluorescent speckles and were used to reconstruct super-resolution images. Using IRIS, Kiuchi et al., convincingly demonstrated improved labeling density, in comparison with conventional immunostaining. Notably, Lifeact could be used for high-density super-resolution imaging in both fixed [8] and living cells [8,85,86].

However, the Lifeact application for actin imaging has several limitations and drawbacks. For example, the Lifeact labeling of filopodia or cofilin-bound actin may be imperfect or fail completely [22,87,88]. In addition, Lifeact exhibits a 10-fold higher affinity to G-actin, leading to background cytosolic actin labeling [14]. Nonetheless, the Lifeact-based imaging of actin is comparable or slightly better than dSTORM (direct STORM) with conventional phalloidin staining [82]. The list of actin-binding probes with low affinity includes Utr261 and F-tractin. The first one is the first 261 amino acid residues of utrophin—the filament-crosslinking protein [89], which binds to F-actin without stabilizing it [90]. With the K_D_ in a micromolar range (18.6 μM [90]), Utr261 can be used for live-cell F-actin imaging [89]. Additionally, the truncated form of Utr261—Utr230—seems to be a unique live-cell probe, which can label short actin filaments in mammalian nuclei [91]. According to a comparative study of Belin et al., another probe—F-tractin (residues 10–52 of rat inositol 1,4,5-triphosphate-3-kinase A [92]) efficiently stains a broad range of actin structures [22]. In addition, the F-tractin probe has the highest exchange rate among all actin tags [22].

Other cytoskeleton structures also could be labeled in a transient manner. For example, microtubule-associated proteins MAP4 and MAP7 interact with microtubules with K_D_ ≈ 0.3 μM [93] and 0.47 μM [94], respectively. MAPs, as well as microtubule plus-end-tracking protein CLIP-170 [95] and tau protein [96], may be used for microtubule labeling [8,86,94,97]. Similarly, intermediate filaments may be labeled with plectin-1 [98]. Of particular note is the motor-PAINT method, based both on PAINT and single-molecule particle tracking [99]. Motor-PAINT utilized the ability of different kinesins to bind to differently oriented microtubules: Kinesin-1 selectively binds to minus-end-oriented microtubules, while Kinesin-3 prefers microtubules that are mostly plus-end-out-oriented. With this technique, one could reveal the orientation of microtubules [100].

## 5. Imaging by Peptide–Peptide Interactions

All the aforementioned methods have limitations in their applicability. Some may only be used in fixed cells, while others require specific probes for different targets. One could envision a universal label, suitable for a multitude of targets and compatible with different microscopy techniques. One of the contenders for the role of a universal labeling agent is K/E-coils. These are short artificial α-helices, which were developed by Chao et al. [101]. Typically, K/E-coils consist of three to five repeating heptads enriched with lysine (K-coils) or glutamate (E-coils) amino acid residues. The heterodimerization of K/E-coils leads to the coiled-coil formation (Figure 4A) and could be fine-tuned in the micromolar-nanomolar range depending on the number of heptad repeats and the amino acid sequence of each heptad (for example, the presence of Ile or Val at the so-called **a** position of a coiled-coil) [102]. While homodimerization is possible, different charges of K/E-coils determine the predominance of heterodimerization over homodimerization.

Originally, K/E-coils were used in affinity chromatography and biosensor applications [103]. Later, K/E-coils and other coiled-coil-forming peptides became well-known dimerization agents for protein labeling. Therefore, Yano et al., used K/E-coils of 3–4 heptad length ((KIAALKE)_3/4_ for K-coils and (EIAALEK)_3/4_ for E-coils) for transient labeling of surface-exposed receptors of living cells [104]. E-coil was attached to an extracellular terminus of various receptors in this work, while K-coil-conjugate with chemical dye was added to the medium. Others used synthetic coiled-coils called SYNZIPs [105,106], but the general principle remained the same [107]. The listed methods demonstrated the highly specific labeling of different proteins in living cells. However, all of the proteins were transmembrane with extracellular parts, and none of the intracellular proteins was labeled.

The first implementation of K/E-coils for labeling proteins inside the cells with different kinds of microscopies, including localization microscopy, was a KECs (K/E-coils) approach [21]. One coil was fused to a target protein, while the other carried fluorescent protein. Thus, the whole system was fully genetically encoded (Figure 4B). Several different compartments were labeled using a set of K/E-coils combinations with varying affinity (membrane proteins caveolin-1 and clathrin, actin, myosin, vimentin, histone H2B, and others). In addition, in partial illumination conditions, such as in TIRF, the photostability of labeling was significantly increased compared to covalent labeling. This feature allowed long-term imaging both in wide-field and localization microscopy conditions. Additionally, the authors demonstrated the usability of KECs for imaging de novo synthesized proteins. Nevertheless, noticeable background in the cytoplasm derived from unbound labels should be mentioned as a disadvantage of KECs labeling.

Later, Eklund et al., combined the imaging principle of DNA-PAINT and K/E-coils as the labeling agent in the peptide-PAINT method [108]. They used two pairs of K/E-coils with K_D_ of 1.7 μM and 81 nM. While one coil was bound to target protein via antibodies, the second was conjugated with Cy3B and freely diffused in the medium (Figure 4C). However, despite the significant decrease in both 𝝉_d_ and 𝝉_b_ comparing to classical DNA-PAINT, it still required antibodies for protein tagging. Therefore, peptide-PAINT is a relatively expensive method that could be used in fixed cells only.

A fully genetically encoded variant of peptide-PAINT was demonstrated by Oi et al. [109]. In this method named LIVE-PAINT, the interaction of SYNZIP17–SYNZIP18 [106] peptides (K_D_ = 1 nM) or TRAP4–MEEVF [110] protein-peptide (K_D_ = 300 nM) were applied for labeling. Using these dimerizing agents and fluorescent proteins as reporters, authors successfully imaged structures in living yeast cells. However, to achieve PAINT conditions, the concentration of labels was very low, resulting in a small number of localizations.

In our opinion, the use of interacting peptides is a very promising labeling approach that may be used both with fixed and living cells. The extension of KECs for multitarget imaging can be achieved, in principle, with orthogonal variants of K/E-coils. The feasibility of multitarget labeling with orthogonal coils was demonstrated with SYNZIP coils labeling of extracellular membrane proteins [107]. Multiple orthogonal coiled-coils were recently reported [111,112,113] and used, for example, in drug delivery systems [114], paving the way to future implementations of multitarget labeling.

## 6. Exchange-STED

Another super-resolution technique that may use the advantages of exchangeable labeling is STED microscopy [115]. In the STED microscope, in addition to the excitation laser beam, a red-shifted high-power STED-laser beam coincides with the excitation laser at the focal plane and depletes fluorescence in the outer region of the PSF by stimulated emission. In the simplest scenario, the STED-laser is engineered to acquire a donut-shaped structure at the focal plane. The stimulated emission of the fluorophores in the outer rim of the donut shrinks the effective PSF to the area near its center, increasing the resolution [43,115]. Theoretically, with the increase of STED-laser intensity, one can reach extremely high resolution [116]. However, in practice, STED-laser power is limited by photodamage of a sample and photostability of labeling.

Recent papers demonstrated the usability of transient labels for STED [74,117,118,119]. In contrast with nanomolar concentrations for PAINT labels, much higher concentrations (≈100 nM–1 μM) were used for exchange-based STED [117]. In addition, the optimal affinity required for fast replacement of the fluorophores in exchange-based STED lies in the range of 1–10 μM [117]. The set of tags that satisfy these requirements include Lifeact (K_D_ = 2.2 µM [14]) and SiR-Hoechst (K_D_ = 8.4 µM [120]) for staining of actin filaments and DNA, respectively [117]. Importantly, the dynamics of target structures in living cells could be registered with STED-enabled high-resolution with such exchangeable probes [119]. Similarly, rapid exchange of fluorogens in the protein-PAINT method provides an improvement in photostability in STED imaging [74].

Another way of performing STED imaging with exchangeable probes is using the Exchange-PAINT [37] (a DNA-PAINT [30] variant for multitarget imaging) labeling technique. Despite several attempts to combine DNA-PAINT with STED [121,122], only Spahn et al., demonstrated improved labeling photostability [118]. In this work, they tuned docking and imager strands to achieve a fast exchange rate. Multitarget (2–4 protein structures) labeling was also demonstrated, achieved by either repeated imaging–washing cycles and orthogonal docking–imager pairs [121,122], or simultaneously staining all targets with orthogonal pairs of strands [118]. Similarly, but using distinct probes for different structures, the dual-color STED in living cells was performed [117].

## 7. Conclusions and Perspectives

Transient labeling, which started with just a few low-affinity tags, has now developed into a pleiad of methods compatible with most modern modalities of fluorescence microscopy (Table 2). Today, transient labels can be used to stain nearly all biomolecules of living cells: proteins, lipids, and DNA.

Importantly, transient labeling is intrinsically well-suited for multiplex high-content imaging due to an easy sequential staining and washing. Notably, not only eukaryotic cells but also bacterial cells were successfully imaged with PAINT [123]. Existing low-affinity labeling methods are compatible with different microscopy setups, ranging from common wide-field and TIRF microscopy to lattice light-sheet microscopy [124].

Since the demonstration of the effectiveness of transient labels for most cellular targets has already been shown, significant progress can be expected in the quality and color palette of these molecular tools. A promising direction is a development of SiR-actin/SiR-tubulin-like fluorogenic dyes [19] but with low-affinity binding. This would pave the way for tracking native cellular proteins with minimal disturbance of target protein functioning due to transient interactions with a dye and absence of a bulky protein tag.

Above all, the versatility concerning target molecules should be improved. Studies need to focus on developing a more common way of staining protein structures, lipid membranes, or nucleic acids with the same or a slightly different approach. In addition, the transient tags with improved and higher signal-to-noise ratio are needed, in order to follow the natural dynamics of cellular structures with minimal photodamage.

## Figures and Tables

**Figure 1 ijms-22-11799-f001:**
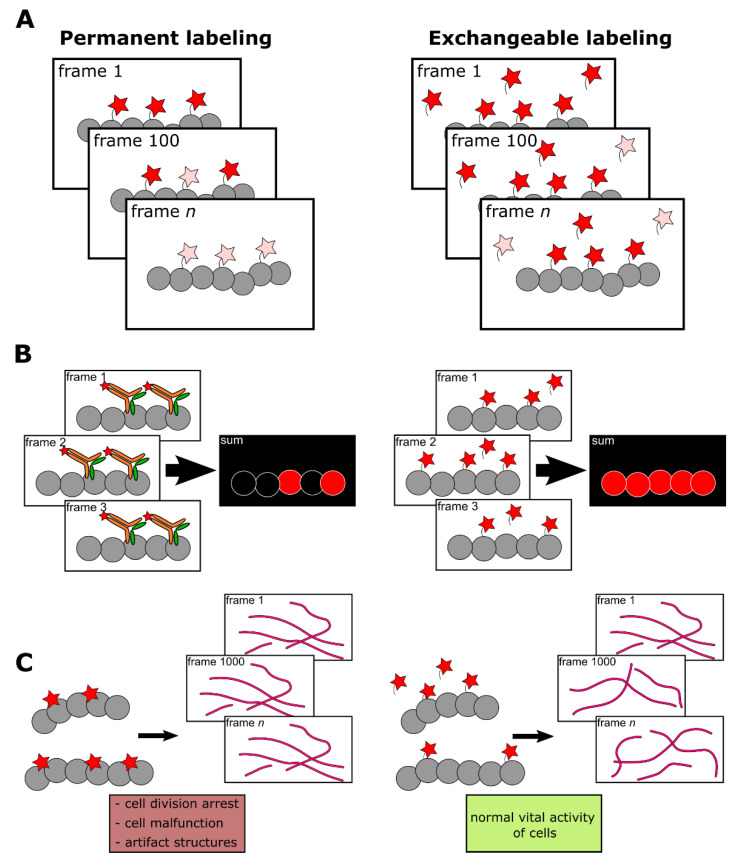
Comparison of permanent (high-affinity or covalent) and exchangeable (transient) labeling. (**A**) Permanent labels are constantly exposed to light irradiation and inevitably photobleached. Therefore, the fluorescent signal deteriorates, making prolonged imaging more complicated. In contrast, the continuous exchange of low-affinity labels with undamaged ones from cytosol or medium increases the apparent photostability of labeling. Red stars denote fluorescent labels, pink stars denote photobleached labels. (**B**) High-affinity labels already bound to target structures, due to their large size, can sterically interfere with the binding of other label molecules. Alternatively, frame-by-frame accumulation of low-affinity labels’ positions followed by frames merging increases effective labeling density. (**C**) Bulky labels, continuously bound to target structures, may affect the dynamics and functioning of the latter. In addition, some labels, such as fluorescently labeled taxol or fluorescent proteins could drastically disturb cell activity. However, in the case of low-affinity labeling, target molecules remain untagged most of the time and therefore their functioning is less hindered.

**Figure 2 ijms-22-11799-f002:**
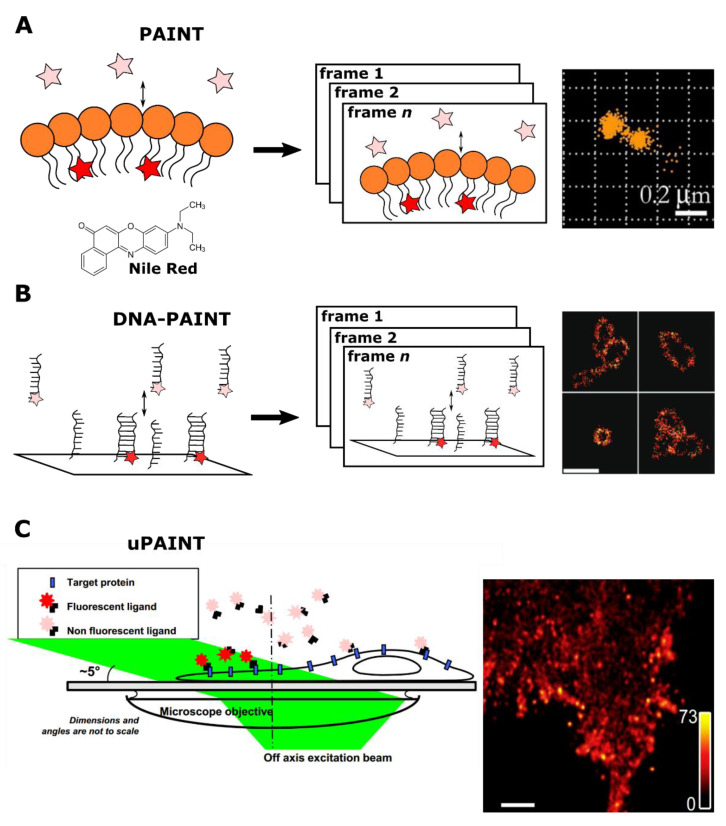
A schematic illustration of PAINT methods. (**A**) An original PAINT approach. Left: an environment-sensitive probe (Nile Red) fluorescently bursts upon reversible interaction with the lipid layer. Center: accumulation of fluorescent bursts during *n* frames registration. Right: the reconstructed image of vesicles, imaged with PAINT approach using Nile Red probe. Reprinted from Sharonov et al. [26] (copyright (2006) National Academy of Sciences). (**B**) A DNA-PAINT approach. Left: in the DNA-PAINT labeling system target and dye molecules are conjugated to complementary DNA strands. Transient interaction between strands temporarily co-localizes fluorescent probes with a target structure. This binding event is detected as a burst of fluorescence. Center: accumulation of fluorescent bursts during *n* frames registration. Right: reconstructed images of long rectangular DNA-origami oligomers labeled with DNA-PAINT. Scale bar 500 nm. Reprinted with permission from Jungmann et al. [30]. Copyright (2010) American Chemical Society. (**C**) A uPAINT approach. Left: fluorescent probes bind to target sites in the cell. A low-angle excitation laser beam (angle of about 5°) illuminates ≈2 μm thick cross section, thereby excites predominantly only bound labels. Right: Super-resolved image of the transmembrane protein TM-6His labeled with trisNTA-AT647N obtained by uPAINT. Scale bar 1 μm. Reprinted from Giannone et al. [31], copyright (2010), with permission from Elsevier.

**Figure 3 ijms-22-11799-f003:**
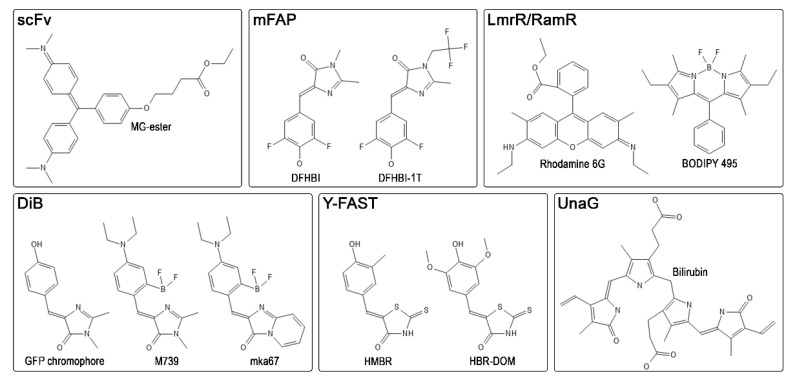
Common representatives of ligands for known FAP-based labeling systems. MG-ester—malachite green ester, DFHBI—difluoro-4-hydroxybenzylidene imidazolinone, DFHBI-1T—3,5-difluoro-4-hydroxybenzylidene-2,2,2-trifluoroethyl imidazolinone, BODIPY—4,4-difluoro-4-bora-3a,4a-diaza-s-indacene, HMBR—4-hydroxy-3-methylbenzylidene rhodanine, HBR-DOM—4-hydroxy-3,5-dimethoxybenzylidene rhodanine.

**Figure 4 ijms-22-11799-f004:**
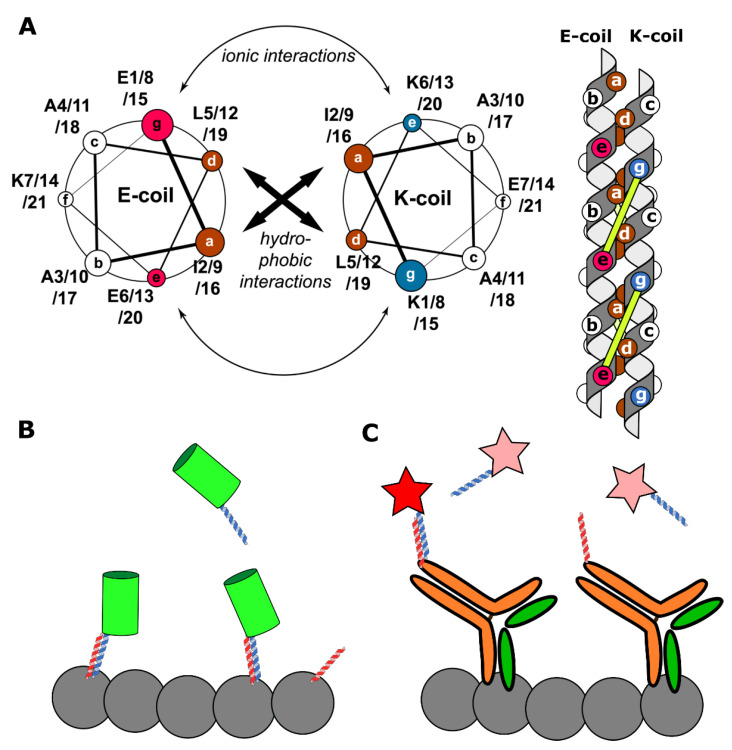
A schematic representation of K/E-coils labeling systems. (**A**) Left: a helical wheel diagram, demonstrating the amino acid residues in interacting helices (reprinted by permission from Springer Nature Customer Service Centre Gmbh: Springer Nature Cellular and Molecular Life Sciences, Perfilov et al. [21], Copyright Clearance Center (2020)). Heptad positions are labeled with **a**–**g**. Brown positions show hydrophobic amino acid residues, purple—positively charged residues, and blue—negatively charged residues. Right: a schematic side view representation of a coiled-coil complex. Circles on helices show amino acid residues from the helical wheel diagram. The complex is stabilized both by hydrophobic interactions between **a** and **d** positions and ionic interactions between residues *i* (position **g**) of one helix and residues *i + 5* of the other helix (position **e**). Ionic interactions are represented by yellow lines. (**B**) The KECs approach. Target proteins (grey circles) are tagged by E-coils (red helices). K-coils (blue helices) are bound to fluorescent proteins (green cylinder). Reversible interaction between K- and E-coils makes possible target proteins visualization. In contrast, in the peptide-PAINT approach (**C**) E-coils are conjugated to target-specific antibodies and K-coils are Cy3B labeled. Similar to the DNA-PAINT, labels are only visible (red star) when a coiled-coil complex is formed, while unbound labels are undetectable (pink stars) [108].

**Table 1 ijms-22-11799-t001:** Characteristics of existing labeling systems based on fluorogen-activating proteins.

System	Protein Size, kDa	K_D_, μM	Color	Super-Resolution Implementation	Reference
scFv	11.2–26.5	0.0012–0.712	blue—far red	SMLM, STED	[17,51,53,81]
UnaG	15.6	0.000098	green	SMLM	[59,61]
Y-FAST	14.0	0.14–16.0	blue—far red	SRRF	[62,65,73]
DiB	18.1	0.1–9.0	green—red	SMLM, STED	[74,75,77]
mFAP	14.0	0.045–11.0	green	not tested	[78,79]
LmrR/RamR	15.0–23.0	0.2–10.0	green—red	not tested	[80]

**Table 2 ijms-22-11799-t002:** Summary of the methods presented in this review.

Method	Target	Super-Resolution Implementation	Fixed/Live-Cell Imaging	Genetically Encoded?	Reference
PAINT	Membranes	SMLM, STED	Both	No	[26,117]
DNA-PAINT	DNA-origami, proteins	SMLM, STED, SOFI	Fixed	No	[30,34,44,118]
uPAINT	Proteins	SMLM	Live-cell	No	[31]
RNA-aptamers	RNA	SMLM	Both	Partially ^1^	[48]
FAPs	Proteins	SMLM, STED, SRRF	Both	Partially ^1^	[17,61,73,77]
IRIS ^2^	Proteins	SMLM, STED	Both	Both ^3^	[8,86,117]
KECs ^4^	Proteins	SMLM	Both	Yes	[21,109]
Peptide-PAINT	DNA-origami, proteins	SMLM	Fixed	No	[108]

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
