# Peer review of "Transient Fluorescence Labeling: Low Affinity—High Benefits"

_ijms, 2021, doi:10.3390/ijms222111799_

Round 1

Reviewer 1 Report

Mishin and co-workers provide a well-written overview on the advancements of fluorescent labelling. A potential title change is suggested to attract a more broad readership, the current title is too specific and it may not attract the level of readership it deserves. The authors need to provide chemical structures when discussing various examples, this includes the fluorescent unit on the GFP, BODIPY, a generic photoswitch, Nile red etc..

I believe once these minor revisions are made the manuscript is now suitable for publication.

Author Response

We thank the Reviewer for in-depth reading of the manuscript and sharing valuable comments and suggestions, which we tried to follow in this revision.

Mishin and co-workers provide a well-written overview on the advancements of fluorescent labelling. A potential title change is suggested to attract a more broad readership, the current title is too specific and it may not attract the level of readership it deserves. 

RESPONSE: We thank the Reviewer for the title change suggestion. Based on previous interactions with the peer scientists in the field, we feel that the concepts of transient, and more importantly, low-affinity labeling are still new for many specialists, and may spark considerable interest in this review. Therefore, we’d like to keep the existing title in order to highlight the focus of the paper. 

The authors need to provide chemical structures when discussing various examples, this includes the fluorescent unit on the GFP, BODIPY, a generic photoswitch, Nile red etc..

I believe once these minor revisions are made the manuscript is now suitable for publication.

RESPONSE: We now added the structure of Nile Red to Figure 2 and include a new Figure 3 with a comparison of GFP chromophore and other molecules mentioned in the manuscript.

Reviewer 2 Report

A well written review paper arguing for the benefits of low affinity fluorescent staining reagents. The paper has a wide coverage but the focus is difficult to identify. The review is very diligent in providing a detail account of ideas and concepts, some of which appear on the periphery of research concerned with or relying on fluorescent labelling.

The manuscript reads well and is likely to generate interest from IJMS readers. One aspect which might require some improvement is the overall focus of the paper. The authors should clearly state and maintain that focus throughout the manuscript.

This review might benefit from a better stricture. The size should be reduced, perhaps by a third. There is no need to cite old references, these do not bring any value to the paper and do not improve it. Some references currently go back to 1958, 1972, 1978 - that is not justified or necessary. About a quarter of the references are older than 10 years.

There is value on this review work, but it might require a substantial effort to make it meet the standards of the IJMS. Whilst it is written in impeccable English, in its current form the paper generates limited impact. 

Author Response

A well written review paper arguing for the benefits of low affinity fluorescent staining reagents. The paper has a wide coverage but the focus is difficult to identify. The review is very diligent in providing a detail account of ideas and concepts, some of which appear on the periphery of research concerned with or relying on fluorescent labelling.

The manuscript reads well and is likely to generate interest from IJMS readers. One aspect which might require some improvement is the overall focus of the paper. The authors should clearly state and maintain that focus throughout the manuscript.

This review might benefit from a better stricture. The size should be reduced, perhaps by a third. 

There is no need to cite old references, these do not bring any value to the paper and do not improve it. Some references currently go back to 1958, 1972, 1978 - that is not justified or necessary. About a quarter of the references are older than 10 years.

RESPONSE: We thank the Reviewer for the comments. We have updated old references to more recent ones wherever possible. However, one of the aims of this review was to show the evolution of transient labeling. Therefore, we cite key papers over 10 years old in order to provide a broader common context to the discussed methods.

There is value on this review work, but it might require a substantial effort to make it meet the standards of the IJMS. Whilst it is written in impeccable English, in its current form the paper generates limited impact. 

RESPONSE: We thank the Reviewer for sharing these comments. We believe that the wide coverage, as indicated by the Reviewer, is the main aim of the current Manuscript. Specifically, we believe that some of the papers cited in this review have not been placed in a proper context, or even omitted in more specialized reviews. We also maintain the emphasis on the low-affinity trait of the discussed probes (quite a counterintuitive concept in the field of fluorescent labeling) throughout the manuscript. With regard to the importance of transient labeling per se, we keep a strong faith in the future of this technology, having tried most, if not all of the discussed labeling methods in our lab. To conclude, we hope that the title “Transient fluorescence labeling: low affinity – high benefits” fully reflects the unique focus of the manuscript. In this revised and restructured version of the paper, we improved the presentation of the data with additional tables and added a small discussion on perspectives and challenges in the field.

Reviewer 3 Report

The review: “Transient fluorescence labeling: low affinity – high benefits” by Perfilov et al. highlights the current status of a wide variety of transiently interacting labels, discusses properties of suitable exchangeable labels, and the advantages of exchangeable probes over permanent ones. Overall, this is an interesting review paper. However, the manuscript can be improved. I recommend this manuscript be accepted with subject to moderate revisions.

Please, take into consideration the following remarks:

  1. There is a lack of appropriate representation through tables. Because it is a review, I think you can summarize and refine some data as a table to make it more readable. For example, some information concerning microscopy techniques and fluorescent probe demands can be organized as a table. Also, it would be advantageous to summarize various methods exploiting exchangeable labels presented in the manuscript as a table.
  2. Can you provide a perspective and challenge section with your one opinion about this field?
  3. I think it is helpful to add an abbreviations section.

Author Response

We thank all the Reviewer for in-depth reading of the manuscript and sharing valuable comments and suggestions, which we tried to follow in this revision.

Please, take into consideration the following remarks:

  • There is a lack of appropriate representation through tables. Because it is a review, I think you can summarize and refine some data as a table to make it more readable. For example, some information concerning microscopy techniques and fluorescent probe demands can be organized as a table. Also, it would be advantageous to summarize various methods exploiting exchangeable labels presented in the manuscript as a table.

RESPONSE: As suggested by the Reviewer, we added two tables to summarize the information about fluorogen-activating proteins and methods mentioned in this Manuscript.

Can you provide a perspective and challenge section with your one opinion about this field?

RESPONSE: We now include a “Conclusions and Perspectives” section, listing some of the challenges and future directions of the field.

I think it is helpful to add an abbreviations section.

RESPONSE: Thank you. We now list all the abbreviations.